# Excessive premature mortality among children with cerebral palsy in rural Uganda: A longitudinal, population-based study

Lukia H. Namaganda[1☉], Rita Almeida[2☉], Dan Kajungu[3], Fred Wabwire-Mangen[4], Stefan Peterson[5,6], Carin Andrews[7], Ann Christin Eliasson[7], Angelina Kakooza-Mwesige[8], Hans Forssberg[7] *

1 Makerere University School of Public Health, Makerere University, Kampala, Uganda, 2 Stockholm University Brain Imaging Center (SUBIC), Stockholm University, Stockholm, Sweden, 3 Iganga Mayuge Health and Demographic Surveillance Site (IMHDSS), Makerere University Centre for Health and Population Research (MUCHAP), Makerere University, Kampala, Uganda, 4 Department of Epidemiology and Biostatistics, Makerere University School of Public Health, Makerere University, Kampala, Uganda, 5 Department of Women's and Children's Health, Uppsala University, Uppsala, Sweden, 6 Department of Global Public Health, Karolinska Institutet, Stockholm, Sweden, 7 Department of Women's and Children's Health, Karolinska Institutet, Stockholm, Sweden, 8 Department of Pediatrics and Child Health, Makerere University College of Health Sciences, Kampala, Uganda

☉ These authors contributed equally to this work.
* hans.forssberg@ki.se

**Data Availability Statement:** The data that underlie the results reported in this article are described at the Swedish National Data Service. Data are made

## Abstract

### Background

Studies from high-income countries reported reduced life expectancy in children with cerebral palsy (CP), while no population-based study has evaluated mortality of children with CP in sub-Saharan Africa. This study aimed to estimate the mortality rate (MR) of children with CP in a rural region of Uganda and identify risk factors and causes of death (CODs).

### Methods and findings

This population-based, longitudinal cohort study was based on data from Iganga-Mayuge Health and Demographic Surveillance System in eastern Uganda. We identified 97 children (aged 2–17 years) with CP in 2015, whom we followed to 2019. They were compared with an age-matched cohort from the general population (n = 41 319). MRs, MR ratios (MRRs), hazard ratios (HRs), and immediate CODs were determined.

MR was 3952 per 100 000 person years (95% CI 2212–6519) in children with CP and 137 per 100 000 person years (95% CI 117–159) in the general population. Standardized MRR was 25·3 in the CP cohort, compared with the general population. In children with CP, risk of death was higher in those with severe gross motor impairments than in those with milder impairments (HR 6·8; p = 0·007) and in those with severe malnutrition than in those less malnourished (HR = 3·7; p = 0·052). MR was higher in females in the CP cohort, with a higher MRR in females (53·0; 95% CI 26·4–106·3) than in males (16·3; 95% CI 7·2–37·2). Age had no significant effect on MR in the CP cohort, but MRR was higher at 10–18 years

available upon request after ensuring compliance with relevant legislation. DOI: 10.5878/xr97-2a37; SND-ID: 2020-178: https://snd.gu.se/en/catalogue/study/preview/b58a1ecd-3f49-4ccd-a855-feab627517b8.

**Funding:** HF received a grant from Swedish Research Council (2017-05474), HF received a grant from the Foundation Frimurare Barnhuset. The funders had no role in study design, data collection and analysis, decision to publish, or preparation of the manuscript.

**Competing interests:** The authors have declared that no competing interests exist.

(39·6; 95% CI 14·2–110·0) than at 2–6 years (21·0; 95% CI 10·2–43·2). Anaemia, malaria, and other infections were the most common CODs in the CP cohort.

## Conclusions

Risk of premature death was excessively high in children with CP in rural sub-Saharan Africa, especially in those with severe motor impairments or malnutrition. While global childhood mortality has significantly decreased during recent decades, this observed excessive mortality is a hidden humanitarian crisis that needs to be addressed.

## Introduction

Global mortality for children below 5 years of age has drastically decreased during the last couple of decades, in keeping with the Millennium Developmental Goals [1,2]. In Uganda, mortality in this age group has declined from 124 deaths per 1000 live births in 2006 to 90 per 1000 live births in 2011 and 64 per 1000 live births in 2016 [3,4]. This decrease has been attributed to antimalarial interventions and breastfeeding, in addition to general socioeconomic development [3]. Nevertheless, child mortality remains high in sub-Saharan Africa [5], especially among children with neurodevelopmental disorders or epilepsy, whose mortality rate was reportedly 3–4 times higher than that of typically developing children [6].

Cerebral palsy (CP) is the most common childhood motor disorder, with a prevalence of approximately 2 per 1000 children in high-income countries (HICs) [7,8] and 3 per 1000 children in low- and middle-income countries (LMICs) [9,10]. CP comprises a range of impairments involving gross, fine and oral motor functions. These impairments can range from mild to severe requiring assistance with all activities, including eating and hygiene. Associated seeing, hearing, communication and cognitive impairments, as well as seizures, are common [8,11].

Several studies from HICs have demonstrated reduced life expectancy of children with CP, especially those with severe motor impairments [12–14]. A study from rural Bangladesh estimated a five-times higher mortality rate in children with CP, compared with the general paediatric population [15]. In our 2015 population-based prevalence study of CP in rural Uganda, we observed a higher prevalence of CP in younger children (4 per 1000 children aged 2–7 years) than in older children (2 per 1000 children aged 8–17 years) [9], suggesting substantial mortality when children reach school age. No population-based study has reported mortality rates of children with CP living in sub-Saharan Africa.

The aim of this study was to examine the mortality rate over a 4-year period of a cohort of children with CP in rural Uganda. We compared this cohort with children of the same age from the general population living in the same area. We hypothesised that the mortality rate would be higher in children with CP than in those without CP and that mortality would be influenced by the severity of CP, nutritional status, age, and sex. We also compared the immediate cause of death (COD) between the two cohorts.

## Methods

### Study design

This was a 54-month longitudinal follow-up study of a population-based cohort of children and adolescents with CP in the Iganga-Mayuge Health and Demographic Surveillance System

(IM-HDSS) in eastern Uganda. This population includes predominantly rural subsistence farmers (80%) living under the poverty income level of 1 US dollar per day. Active surveillance has been performed annually since 2005 by trained fieldworkers who collect demographic data (e.g., pregnancies, births, deaths, migration). IM-HDSS includes one regional referral hospital (Iganga Hospital), 27 health centres, and 132 private health providers (mostly drugstores) and only limited rehabilitation services for children [11,16,17].

## Ethical approval

The study protocol was approved by the Higher Degrees Research and Ethics Committee, School of Public Health, College of Health Sciences, Makerere University and the Uganda National Council for Science and Technology (Reference HS 2608). All caregivers gave written informed consent, and participants provided assent when possible.

## Participants and procedures

The study included two cohorts of children and adolescents aged 2–21 years. The CP cohort consisted of 97 children with CP who were aged 2–17 years when identified through a three-stage screening process in 2015 using the definition of CP used in the Surveillance of CP in Europe [9,18]. The general population cohort comprised all 41 319 children aged 2–17 years living in IM-HDSS on 1 January, 2015, and followed until 31 December, 2017.

Families of children with CP, who had been identified in 2015, were contacted by phone or home visit in September 2019 to inquire whether the child was alive and to obtain consent for participating in the study. If the child was deceased, an experienced IM-HDSS field worker with counselling skills conducted a verbal autopsy audit following standard HDSS procedures [5,19]. Surviving children were subjected to follow-up assessments, which will be reported elsewhere.

To analyse mortality risk factors, including impairment severity and nutrition status, we used assessments performed at the first screening session in 2015 [11]. Gross motor impairments were assessed by a team of expert therapists using the Gross Motor Function Classification System (GMFCS) [20], which classifies ability on a five-point ordinal scale, from I (mild/independent) to V (severe limitations requiring assistance for all activities). Information on associated visual, hearing, intellectual, and behaviour impairments and the presence of seizures was gathered from three sources: (1) interview with a study nurse, (2) clinical examination by a clinical officer/paediatrician; and (3) assessment by a therapist, including the Ugandan version of the Pediatric Evaluation of Disability Inventory [21,22]. Impairments were considered confirmed when reported from at least two sources. Weight and height of each child were measured using World Health Organization (WHO) growth standards [23]. Data were missing for GMFCS and associated impairments/seizures in three deceased children and for weight and height in one deceased child. We excluded for each analysis the cases for which we had no data.

For children in the general population, information was collected from IM-HDSS annual censuses. For details see [5]. We identified deceased children and recorded their dates of birth and death and immediate COD.

## Analysis

The primary outcome was all-cause mortality during the 54-month follow-up period (1 March, 2015–30 August, 2019) for children with CP. Person observation months were calculated. Individuals alive or migrated from IM-HDSS before the end of follow-up were right censored [6]. For children in the general population cohort, person observation months were

calculated from 1 January, 2015, until date of death or 31 December, 2017. Children alive or migrated before 31 December, 2017, were assigned 36 months of observation.

To analyse the effect of the level of motor impairments on mortality, children with CP were divided into three GMFCS subgroups: mild (Levels I–II), moderate (Level III), and severe (Levels IV–V). To evaluate associated impairments/seizures, the children were divided into two groups: one or fewer impairments/seizures and more than one impairments/seizures. Nutritional status was assessed by calculating Z scores for weight-for-age (WFA), height-for-age (HFA), and weight-for-height (WFH). Children were divided into two categories: severe malnutrition (Z score below –3 standard deviations (SD) for at least one of the three measures, according to age and sex) [24] and no severe malnutrition (Z score above –3 SD for all three measures).

Immediate COD was determined from information obtained from verbal autopsy question-naires derived from WHO, according to standard HDSS operation procedures validated in Uganda [19,25]. Death was monitored through biennial home visits and notifications of com-munity based scouts. Specially trained field workers with counselling skills performed a verbal autopsy after allowing the family four to six weeks of mourning. Two certified physicians assigned by HDSS independently reviewed the information and assigned COD using the inter-national classification of death algorithm (ICD-10 codes). COD was confirmed if both physi-cians were in agreement; otherwise, a third physician was consulted.

Data are presented as mean ± standard deviation unless otherwise indicated. We double-entered and verified data using Epi-data Entry-V3.1. R software (version 3.6.3.; epitools, sur-vival, and survimer packages) for statistical analysis [26–29].

Mortality rates (MRs), i.e., the frequency of death occurrence, and corresponding 95% con-fidence intervals (CIs) per 100 000 person observation years were calculated for groups and subgroups. Confidence intervals were estimated assuming that number of events followed a Poisson distribution. MR ratios (MRRs) and corresponding 95% CIs were calculated to com-pare mortality between the CP cohort and general population. Standardized MR and standard-ised MRR were calculated with general population as reference. The standardised measures take into account the age structure of the two populations in order to calculate MR and MRR in case the CP cohort would have a different age structure than the reference population.

Cox proportional hazard regression models were fitted to compute hazard ratios (HRs) and investigate in separate models the contributions of age, sex, GMFCS, associated impairments/ seizures, and severe malnutrition on all cause-mortality among children with CP. For com-pleteness we also investigated the contribution of weight-for-age (WFA), height-for-age (HFA), and weight-for-height (WFH) in separate models. GMFCS subgroup, associated impairments/seizures, and severe malnutrition were also included in a single multivariable Cox proportional hazard regression model. The proportional hazards assumption was verified for all models.

Survival probabilities were estimated using the Kaplan-Meier method. Survival curves were compared between groups and subgroups using the log-rank test.

## Results

All 97 children with CP who were examined 2015 were identified 2019. Fifteen children had died during the study period, at a mean age of 10·2±5·9 years. Of the children who died, 9 had GMFCS III–V; 8 had more than one associated impairments/seizures; and 10 had severe mal-nutrition. Of the 41 319 general population children, 169 died during their 3-year observation period. The mean age at death was 7·2±4·8 years. Survival probabilities for both cohorts are shown in Fig 1A.

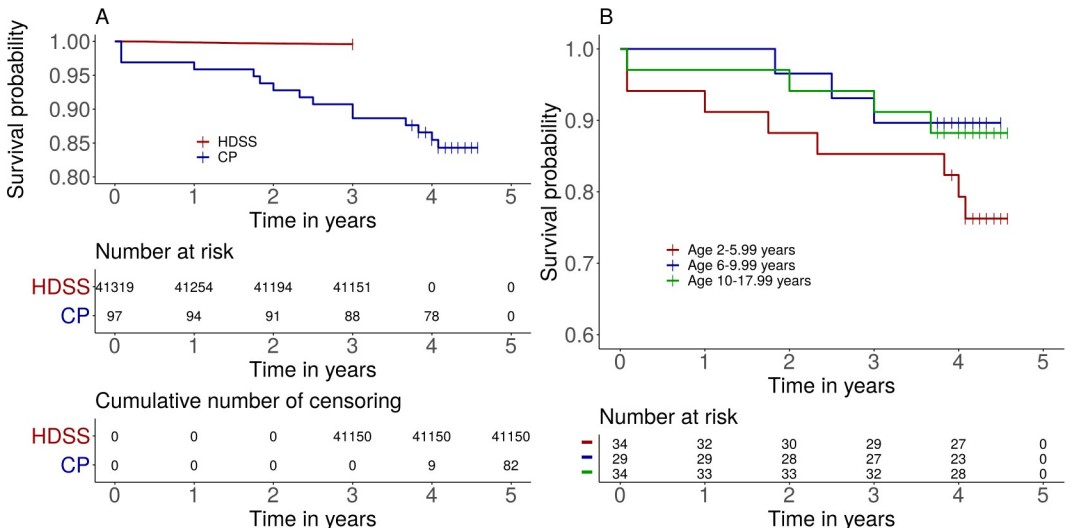

**Fig 1.** A) Estimated survival probability for children with cerebral palsy and age matched children of the general population living in the IM-HDSS (log-rank test **p<0.0001**). Number at risk and cumulative number of censored by time are shown in the tables. B) Estimated survival probability of children with cerebral palsy comparing three age groups: 2–5 years; 6–9 years; and 10–17 years (**p = 0·28**). The short vertical lines indicate censored data.

## Mortality rate and mortality rate ratio

MRs for the two cohorts and the MRRs are shown in Table 1. MRR of the CP cohort was 29·0 (95% CI 17·1–49·1; p<0·0001), with general population as the reference. The standardized MR was 3455 per 100 000 person years for the CP cohort, with a standardized MRR of 25·3.

## Age

In the general population, MR was 298 per 100 000 person years in the youngest age group (2–6 years) and 75 per 100 000 person years in the oldest group (10–18 years). The effect of age was significant in the general population (log-rank, p<0·0001). In children with CP, MR was 6255 per 100 000 person years in the youngest group (2–6 years), compared with 2575 and 2960 per 100 000 person years in the older age groups (6–10 years and 10–18 years, respectively). Survival probability was not significantly different between age groups in the children with CP (log-rank, p = 0.28; Fig 1B). MRR for children with CP was 21·0 for the youngest

**Table 1. Age- and sex-specific mortality rates and mortality rate ratios for the cerebral palsy cohort and general population.**

| Age (y) and sex | Cerebral palsy cohort | | | General population | | | Mortality rate ratio (95% CI) |
|---|---|---|---|---|---|---|---|
| | Deaths (n) | Person years | Mortality rate per 100 000 person years (95% CI) | Deaths (n) | Person years | Mortality rate per 100 000 person years (95% CI) | |
| 2·0–5·9 | 8 | 128 | **6255** (2701–12 326) | 93 | 31 247 | **298** (240–364) | **21.0** (10·2–43·2) |
| 6·0–9·9 | 3 | 116 | **2575** (531–7526) | 31 | 32 324 | **96** (65–136) | **27.0** (8·2–88·2) |
| 10·0–17·9 | 4 | 135 | **2960** (806–7579) | 45 | 60 113 | **75** (55–100) | **39.6** (14·2–110·0) |
| Male | 6 | 221 | **2712** (995–5904) | 102 | 61 348 | **166** (135–201) | **16.3** (7·2–37·2) |
| Female | 9 | 158 | **5685** (2600–10 792) | 67 | 62 335 | **107** (83–137) | **53.0** (26·4–106·3) |
| Total | 15 | 380 | **3952** (2212–6519 | 169 | 123 684 | **137** (117–159) | **29.0** (17·1–49·1) |

CI = confidence interval; n = number; y = years.

group and 39·6 for the oldest group. Thus, although MR decreased (albeit not significantly) with increasing age in the CP cohort, MMR increased with age because of the steeper decrease in MR with increasing age in the general population.

## Sex

In the general population, MR was significantly higher in males (166 per 100 000 person years) than in females (107 per 100 000 person years; p = 0.009; Table 1). Conversely, in the CP cohort, MR was higher in females (5685 per 100 000 person years) than in males (2712 per 100 000 person years), although there was no significant sex effect on survival probability (log-rank, p = 0·15; Fig 2A). When we used a Cox proportional hazard regression model including the CP and general population groups, we observed significant interaction between sex and group (p = 0·032), indicating that the effect of sex differed between the CP and general population cohorts.

## Motor impairment

We evaluated the association between MR and severity of motor impairments, as well as malnutrition, within the CP group (Table 2). MR was higher in children with severe motor impairments (GFMCS IV-V; 8718 deaths per 100 000 person years) than in those with mild impairments (GMFCS I-II; 1305 deaths per 100 000 person years). There was a significant effect of GMFCS on estimated survival probability test (log-rank, p = 0·009; Fig 2B). In Cox model analysis, using GFMCS alone as the explanatory variable, risk of death was almost seven

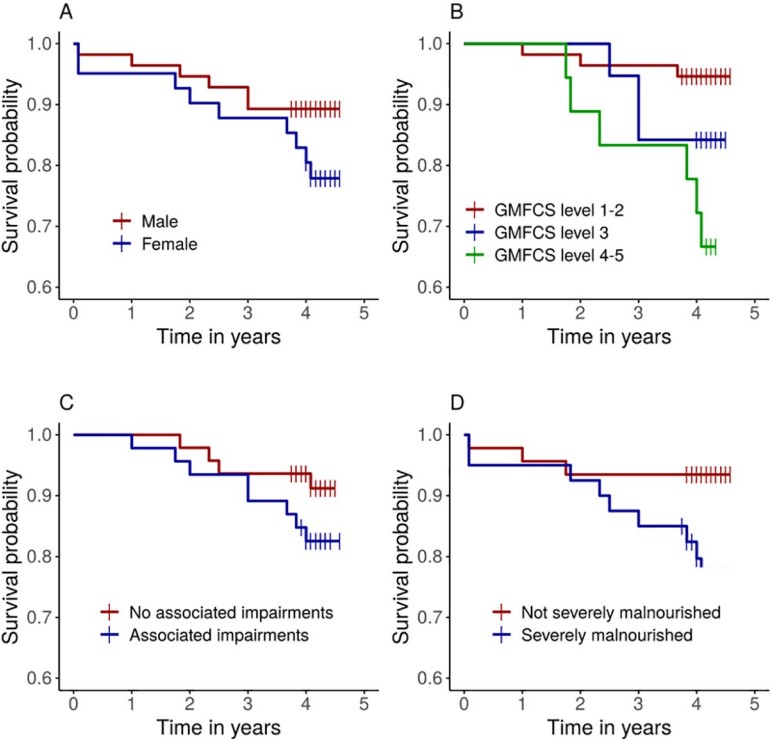

**Fig 2.** Estimated survival probability according to A) sex (log-rank test **p = 0·15**); B) GMFCS (**p = 0.009**); C) Associated impairments (**p = 0·213**); D) Severe malnourishment (**p = 0·037**). The short vertical lines indicate censored data.

**Table 2. Mortality rates of children with cerebral palsy according to severity of motor impairments and presence of associated impairments/seizures or severe malnutrition.**

| | Deaths (n = 15) | Person years | Mortality rate per 100 000 person years | P-value |
|---|---|---|---|---|
| **GMFCS level** | | | | 0·009[a] |
| I-II (mild) | 3 | 230 | **1305** (269–3813) | |
| III (moderate) | 3 | 77 | **3913** (807–11437) | |
| IV-V (severe) | 6 | 69 | **8718** (3199–18 976) | |
| Not classified | 3 | | | |
| **Associated impairments/seizures** | | | | 0·213 |
| One or less | 4 | 19 232 | **2080** (567–5325) | |
| More than one | 8 | 18 312 | **4369** (1886–8608) | |
| Not classified | 3 | | | |
| **Malnutrition** | | | | 0·037 |
| Not severe[b] | 3 | 18 455 | **1626** (335–4751) | |
| Severe[b] | 9 | 15 305 | **5880** (2689–11 163) | |
| Not assessed | 3 | | | |

GMFCS = Gross Motor Function Classification System; n = number. P values were calculated with the log-rank test comparing two categories.

[a] With GMFCS level I–II as the reference.

[b] Severe malnutrition was defined as a Z score below –3 standard deviations (SD) for at least one of three measures (weight-for-age, height-for-age, and weight-for-height). Not severe malnutrition was defined as a Z score above –3 SD for all three measures.

times higher in children with severe motor impairments than in those with mild impairments (HR 6·8; p = 0·007; Table 3). The difference in risk between mild and moderate impairments was not significant (p = 0·178).

## Associated impairments

MR was higher in children with more than one associated impairments/seizures than in those with one or fewer impairments (Table 2). However, survival probability was not significantly related to the number of associated impairments (log-rank, p = 0·213; Table 2, Fig 2).

## Malnutrition

Children with severe malnutrition had higher MR than children without severe malnutrition (Table 2). Survival probability was significantly lower in children with severe malnutrition than in those without (log-rank, p = 0.037; Fig 2D). Risk of death was almost four times higher in children with severe malnutrition than in those without severe malnutrition (HR, 3·7; p = 0·052; Table 3). When using WFA, HFA and WFH as explanatory variables in three separate models we found that only weight-for-age (WFA) had a significant effect (p = 0.023).

**Table 3. Univariable and multivariable analysis of risk factors for mortality.** For univariable analysis, separate models were used for GMFCS subgroups, associated impairments/seizures, and severe malnutrition. These parameters were also included in one single multivariable model.

| Risk factors | Univariable HR (95% CI) | P-value | Multivariable HR (95% CI) | P-value |
|---|---|---|---|---|
| GMFCS | | | | |
| Level III | 3·0 (1·3–14·9) | 0·178 | 4·3 (0·3–56·7) | 0·265 |
| Level IV-V | 6·8 (1·7–27·3) | 0·007 | 15·4 (1·5–156·3) | 0·021 |
| Associated impairments/seizures | 2·1 (0·6–7·0) | 0·223 | 1·2 (0·3–4·8) | 0·795 |
| Severe malnutrition | 3·7 (1·0–13·5) | 0·052 | 1·7 (0·3–10·0) | 0·560 |

CI = confidence interval; GMFCS = Gross Motor Function Classification System; HR = hazard ratio.

In a multivariable model including GMFCS subgroup, associated impairments/seizures, and severe malnutrition, only GMFCS was significantly associated with an increased risk of death, with a HR of 15·4 (95% CI 1·5–156·3; p = 0·021) for children with severe impairments, compared with those with mild impairments (Table 3). In this model, we did not include age and sex since these variables did not have a significant effect in the CP population in previously described analysis. To further understand the results of the multivariate model we tested for independence between the three variables using the Chi-square test of independence. We found evidence for an association between GMFCS and severe malnutrition (p<0.0001), but not between any other variables.

## Immediate cause of death

Immediate CODs of the CP- and HDSS populations are shown in Table 4 (See also S1 Table). In the children with CP anaemia/malnutrition (n = 6; 40%), malaria (n = 4; 33%), pneumonia (n = 3; 17%), and meningitis (n = 2; 13%) were the immediate COD. Children dying from anaemia had malaria as an underlying cause (See S1 Table). In the general population, anaemia/malnutrition (22%), malaria (20%), gastro-intestinal disorders or diarrhoea (13%), non-communicable diseases (18%), and external causes including injuries (10%) were the most common COD. COD could not be determined in 38 children because caregivers with information about the deceased, who were required to perform the verbal autopsy, had moved to unknown locations (n = 18) or were not at home to complete the interviews (n = 19) or it was not possible to determine COD (n = 1).

## Discussion

In this study, we demonstrated that children with CP living in a rural region of eastern Uganda have excessive premature mortality, with a 25-times higher risk of death, compared with general population children of the same age in the same region. In children with CP, the risk of death was particularly high in those with severe motor impairments and severe malnutrition, and females and older children tended to have higher relative risk of death, compared with the general population.

The 25-fold higher MR in children with CP is extreme and contrasts with the results of studies in HICs showing only modestly reduced life expectancy in children and adolescents with CP [30]. Two recent studies showed increased MRs in children with neurological

**Table 4. The immediate COD of children with CP (2[nd] column) and children of the general population at the IM-HDSS (3[rd] column) expressed in percentage.** Note, that none of the children with CP complied to any of the COD at the bottom of the table.

| Cause of Death | CP[a] N = 15 (100%) | HDSS[b] N = 131(100%) |
|---|---|---|
| Anaemia/malnutrition | 6 (40) | 29 (22) |
| Other infections | 5 (33) | 7 (5) |
| Malaria | 4 (27) | 25 (19) |
| GI/Diarrhoea | | 17 (13) |
| External/injuries | | 13 (10) |
| Tetanus | | 6 (5) |
| CNS/Epilepsy | | 10 (8) |
| Other (non-communicable) | | 24 (18) |

[a] Note that children with CP only matched the three first COD categories.

[b] Note that COD could not be determined for 38 deceased children.

impairments in Kenya [6] and in children with CP in Bangladesh [15]. The Kenyan study reported 3–4 times higher MR in children with neurological impairments who were 6–10 years of age. The authors suggested that MR was likely even higher, since their sensitivity analysis indicated that the rate was underestimated. They also noted that children with cognitive and motor impairments, who likely had CP, were largely responsible for the high mortality. MR of the general population of children aged 6–10 years was similar between the Kenyan study and our study (99.8 vs. 95.9 death per 100 000 person-years), validating the MR of our general population and suggesting similar levels of risk factors in the two countries. Crude MR in children with CP in Bangladesh (1950 deaths per 100 000 person-years) was approximately half the MR observed in our study (3952 deaths per 100 000 person years) [15]. However, the true MR in Bangladesh is probably higher because many children likely died before being diagnosed, as the mean age of diagnosis was 5 years. Nevertheless, the MR among children with CP in Bangladesh was 5 times higher than in the general population, in contrast to 25 times higher in Uganda. This may be at least partly attributed to differences in the reported mortality of general population children between the two countries, with a higher MR in Bangladesh than in Uganda (403 vs. 137 deaths per 100 000 person-years) [15,31]. The excessive premature mortality found in our study might thus be representative of rural areas in sub-Saharan Africa and other LMICs, especially in areas where malaria is endemic, as in Iganga/Mayuge. Approximately 80% of households in this region live under the poverty level. Healthcare and other types of support for children with disabilities are scarce, leading to a high risk of premature death in these children [11]. Similar conditions are prevalent in many sub-Saharan countries, emphasizing the need for more population-based studies to provide evidence of a major humanitarian failure that must be addressed.

We found that children with more severe motor impairments, often in combination with severe malnutrition, had the highest MR. This is consistent with studies from both HICs [12,13] and LMIC [10] reporting that disability severity is the strongest predictor of premature death. In our study, we used GMFCS levels and number of associated impairments/seizures to measure CP severity. Children with severe GMFCS levels (IV–V) had an almost seven-times higher risk of death than children with mild CP (levels I–II). Although MR was two times higher for children with more than one associated impairments/seizures, this did not reach statistical significance. A known risk factor for death reported in most studies is malnutrition [10,12]. We found that the majority of children with CP had some degree of malnutrition (below –1 SD) and noted that 10 of the 15 children who died during follow-up had severe malnutrition, with Z-scores below –3 SD for at least one of the three assessed measures (WFA, HFA, and WFH). Children with severe malnutrition had a more than three-times higher risk of death than children without severe malnutrition. Children with severe impairments cannot feed independently and often have impaired oral motor function, leading to chewing and swallowing problems and the need for special foods and prolonged feeding times. They also have an increased risk of aspirating food, resulting in respiratory difficulties and infections. In the CP cohort from Bangladesh, nearly half of the children who died had swallowing difficulties; most of whom died from respiratory or other infections [15]. In HICs, there has been a trend towards longer survival of severely impaired children with CP [12,14] because of improved feeding practices, including tube feedings and gastrostomies. Although these interventions could be introduced in low-resource settings, they are associated with potential complications that could lead to substantial morbidity and even death if not managed appropriately. Thus, there is a need to develop simpler, culturally adapted strategies to address malnutrition among children with CP in LMICs.

We found that girls tended to have a higher MR than boys in the CP cohort; this differed from the general population in which males had a significantly higher MR than females. This

reverse relationship was reported in a Swedish CP cohort, in which females had a higher risk of death than males [13], whereas a study from the United States found that among people with CP who were able to walk, life expectancy was higher in females [14]. Sex was not significantly associated with mortality among children with neurological impairments in Kenya, but the trend was similar to our results: higher MR among females in the impaired group and higher MR among males in the general population [6]. The higher mortality among girls suggests that they may receive different care than boys [32]. Almost no research exists regarding maltreatment of children with disabilities in East Africa [33], but studies from HICs showed that both girls and boys are at high risk of neglect and abuse.

Our results also revealed that MRR was higher in older age groups. This was consistent with our previous findings in this cohort of children with CP, which showed a notable decrease in number of children within the cohort at approximately 8 years of age [9]. These findings suggest that the age pattern for mortality in children with CP differs from that of the general population, and that many died when approaching school age. This suggestion is supported by the older mean age of death in the CP cohort (10·2 years), compared with the general population (7·2 years), and the three-times higher risk of death in the youngest age group of the general population but not of the CP cohort. The Bangladesh study reported that children with CP under age 5 years had the highest risk of death [10], but the results were not compared with the general population, so they may simply reflect the high under-5 years MR of all children. Our relatively high number of children dying after age 6 years may be attributed to caregivers eventually losing hope when they realize that their child will not be cured [11]. Typically developing children are increasingly more independent but children with severe CP are not. Caregivers face many challenges, such as stigma and negative attitudes, when caring for their child and often lack knowledge of their child's condition or prognosis and are fearful of the future [11,34–36]. Additionally, caregivers are typically overburdened with numerous tasks, including household chores, farming, and caring for other children. They may gradually spend less time with their child with CP (including during supervised feedings), increasing the child's vulnerability as he or she grows older.

The verbal autopsies revealed that children with CP were succumbing to anaemia/malnutrition, malaria or other infections. Malaria is endemic in the area and malaria and anaemia/malnutrition were also the most common CODs in the general population, while half of the children in the general population had other CODs such as gastrointestinal disorders and injuries. In a previous study on the under-10-mortality in the IM-HDSS in 2005–2015, the investigators described a shift of COD between 0–5 and 5–9 years, in which gastrointestinal disorders and injuries emerged after 4 years of age [5]. They suggested this resulted from new behaviours, curiosity and limited supervision exposing the children to injuries such as poisoning, falls, drowning and suffocation, or to ingestion of unhealthy items, such as worms or typhoid, leading to diarrhoea and gastrointestinal disorders. These CODs were not seen in the CP cohort since deceased children were unable to walk. Instead, they retained a similar COD pattern as children below 5 years in the general population, i.e., malaria, malnutrition and infections [5]. In regions where these risks are prevalent, for example the endemic spread of Malaria, children with CP are more vulnerable and more likely to die. The high incidence of anaemia as COD likely reflects a combination of underlying malaria infections and severe malnutrition in severely impaired children with feeding problems. Currently, no support is provided to these children and their families in Iganga/Mayuge [11]. Providing simple preventive measures, such as use of insecticide-treated mosquito nets to prevent malaria infections, or devising innovative ways to supply simple, ready-to-administer, locally available nutritious foods coupled with caregiver training and support would likely produce considerable improvements in nutritional status and increased resistance to infection [3,37,38].

## Strengths and limitations

There are enormous challenges studying the epidemiology of children with CP in rural sub-Saharan Africa such as lack of population registers, poorly enacted laws and policies regarding their specific needs and rights, stigma and marginalization of children with disabilities [34–36]. The unique strength of this study was the utilization of the IM-HDSS, which performs annual surveys registering child birth and child death and with a validated system determining the COD through verbal autopsies. The infrastructure of the IM-HDSS allowed us to perform a three stage screening of all children living in the area in 2015. However, due to the stigma there was a risk that children with CP were hidden by their families and not reported in the screening. We took several measures in the screening to find all children with suspect symptoms, including adapting screening questions to the local culture and terminology, and a triangulation by village key informants to identify cases missed in the screening (for details; [9]). Yet, there was a risk that some children with CP were missed in the 2015 screening, which would also affect the results of this study. Note also that children dying prior to two years of age were not included in this study. Another limitation is the relatively small number of children with CP, which reduced the statistical power and did not allow more detailed analyses.

The usage of contemporary diagnostic and classification systems and assessment tools for cerebral palsy, administered by an experienced team of trained clinicians, is a strength making it possible to compare the results with studies from other countries. The COD of the deceased children was determined through verbal autopsies, i.e., interviews with the caregivers. Although these interviews were made within some months according to the IM-HDSS procedure, there was a risk of information and recall bias. The COD of anaemia and malaria, were not set by blood counts or malaria diagnostics, but based on information from the caregiver of the deceased child, sometimes this information had been achieved from visits at health centres. Finally, this study was performed in a rural geographic area where malaria is endemic, which limits the generalisability of our results to more urban areas or those with less widespread malaria.

## Conclusion

Our findings provide evidence of excessive premature mortality among children with cerebral palsy in sub-Saharan Africa, especially in those with severe motor impairments or malnutrition. Females and older children with CP had higher relative risks of death in relation to the general population. While global childhood mortality has significantly decreased during recent decades, this observed excessive mortality is a hidden humanitarian crisis that needs be addressed. Research on mortality and risk factors in children with developmental disabilities is required in low- and middle-income countries where there is a lack of population based information. In order to develop strategies to monitor and combat mortality in children with disabilities, there is a need to identify long-term risk factors and immediate cause of deaths. Furthermore, existing laws and policies related to their specific requirements need to be reinforced. In addition of recognizing severe gross motor impairments and severe malnutrition as strong risk factors in our cohort of children with CP, verbal autopsies identified malnutrition, anaemia and as most likely COD. These results can guide efforts to develop preventive measures to reduce mortality in this vulnerable group, e.g., insecticide-treated mosquito nets to prevent malaria infections, and targeting malnutrition by providing locally available nutritious foods coupled with caregiver information and support.

## Supporting information

**S1 STROBE checklist. STROBE 2007 (v4) statement—checklist of items that should be included in reports of *cohort studies*.**
(DOCX)

**S1 Table. Immediate COD and risk factors for the 15 deceased children with cerebral palsy.** NA = not available data; 3 children did not complete functional assessments, and one child had incomplete anthropometric measurements at first examination. Note that 6 children had chronic malaria infections as underlying COD. Sign "–"indicates no underlying COD. (DOCX)

## Acknowledgments

We thank the project coordinator Keron Ssebyala, the clinicians performing the assessments, and the management of the Iganga/Mayuge Health Demographic Surveillance System.

## Author Contributions

**Conceptualization:** Angelina Kakooza-Mwesige, Hans Forssberg.

**Data curation:** Lukia H. Namaganda, Rita Almeida, Dan Kajungu, Hans Forssberg.

**Formal analysis:** Lukia H. Namaganda, Rita Almeida, Dan Kajungu, Hans Forssberg.

**Funding acquisition:** Hans Forssberg.

**Investigation:** Lukia H. Namaganda, Dan Kajungu, Carin Andrews.

**Methodology:** Rita Almeida, Angelina Kakooza-Mwesige, Hans Forssberg.

**Project administration:** Lukia H. Namaganda, Angelina Kakooza-Mwesige, Hans Forssberg.

**Resources:** Hans Forssberg.

**Software:** Rita Almeida.

**Supervision:** Lukia H. Namaganda, Fred Wabwire-Mangen, Angelina Kakooza-Mwesige, Hans Forssberg.

**Validation:** Rita Almeida, Stefan Peterson, Angelina Kakooza-Mwesige, Hans Forssberg.

**Writing – original draft:** Hans Forssberg.

**Writing – review & editing:** Lukia H. Namaganda, Rita Almeida, Dan Kajungu, Fred Wabwire-Mangen, Stefan Peterson, Carin Andrews, Ann Christin Eliasson, Angelina Kakooza-Mwesige.

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
