## [Decision Letter · Decision Letter 0]

4 Nov 2020

PONE-D-20-31617

Excessive premature mortality among children with cerebral palsy in rural Uganda: a longitudinal, population-based study

PLOS ONE

Dear Dr. Forssberg,

Thank you for submitting your manuscript to PLOS ONE. After careful consideration, we feel that it has merit but does not fully meet PLOS ONE’s publication criteria as it currently stands. Therefore, we invite you to submit a revised version of the manuscript that addresses the points raised during the review process.

Note that you have indicated that all data of your study are available, but these have not been provided in the manuscript and supplementary material as indicated. You would need either to provide a link to the location where data can be downloaded, or, if for ethical or other reasons this is not possible, explain the reasons for not publishing the full dataset. See details at https://journals.plos.org/plosone/s/data-availability for Plos One's policy.

Please use the STROBE checklist for reporting of observational study, and consider addressing the limitations of  observational studies.

We look forward to receiving your revised manuscript.

Kind regards,

Barbara Schumann, Ph.D.

Academic Editor

PLOS ONE

Journal Requirements:

2. Please remove your figures from within your manuscript file, leaving only the individual TIFF/EPS image files, uploaded separately.  These will be automatically included in the reviewers’ PDF.

Reviewers' comments:

Reviewer's Responses to Questions

**Comments to the Author**

1. Is the manuscript technically sound, and do the data support the conclusions?

Reviewer #1: Yes

Reviewer #2: Yes

2. Has the statistical analysis been performed appropriately and rigorously? 

Reviewer #1: Yes

Reviewer #2: Yes

3. Have the authors made all data underlying the findings in their manuscript fully available?

Reviewer #1: Yes

Reviewer #2: No

4. Is the manuscript presented in an intelligible fashion and written in standard English?

Reviewer #1: Yes

Reviewer #2: Yes

5. Review Comments to the Author

Reviewer #1: Case ascertainment is the greatest limitation of the study, simply because the cases are too few, indicating there may have been some others left out for follow-up. Stigma for cerebral palsy (CP) is rampant in these rural settings of subs-Saharan, and for this reason many children with CP will be hidden without disclosing the status to the census staff.

Many children will probably already have died by the 2015, when this study was set up, which can result in few cases available for follow-up in this study (N=97). Reliability of these followed numbers can be done by computing a prevalence and examining if it compares with expected estimates from low income countries (e.g. 4 per 1000 previously reported in Uganda).

Failure to identify children with CP increases their inadvertent inclusion into the comparison group selected from the general population, resulting in higher mortality rates in controls, subsequently reducing the comparative or relative mortality ratios.

These low numbers due to failure to exhaustively identify all cases in the population would introduce selection bias, making the interpretation of these results difficult.

The mortality rates will increase with duration since onset of CP, so early identification of CP would reduce the gross underestimation which is possible with birth cohorts and registers.

In addition to the 97 children identified in 2015, the authors could have considered other ascertain methods including hospital facilities, and rehabilitation services for children who visited these facilities in 2015.

The period of follow-up is a relatively short and wouldn’t identify many deaths as would a longer follow up, particularly in the general population, which results in higher mortality ratios when compared with CP cases.

The follow up frequency isn’t very clear in the methods, as to how many follow up phone calls were made over the period for those who were alive on the initial follow up. Multiple sweeps of follow ups would increase the probability of picking up more deaths. This need to be clarified in the methods, otherwise it can complicate the identification of deaths.

Severe motor impairments are on the spectrum and part of CP and would rather be considered for stratified analysis rather than risk factors analysis. Analysis of malnutrition as a risk factor is justified.

How did the individual levels of malnutrition perform in the risk factor analysis i.e. wasting (weight-for-height z scores), weight for age z scores or stunting (height-for-age z scores) compared to the combined category for any of these? These could differentially influence mortality outcomes.

It is worth emphasizing the combined malnutrition wasn’t significant at the multivariate but at the univariable level.

The role of co-existing conditions such as sensori-neural impairments, cognitive impairments and epilepsy are not explicitly reported, yet these can increase premature mortality. If these co-existing conditions are to be examined, they could be separated from risk factors analysis.

Causes of death should compared between those for CP and those for the general population.

The older mean age of death for CP compared to controls needed further discussion, as this could suggest continuing risk of death for CP, or that underlying causes of death are different between these groups. Non-infectious causes of death may continue to pause risk in even older ages, while most infections e.g. malaria are in early years.

More limitations need to be noted in the discussion including some outlined above.

Discretionary comments

Ensure the formatting instructions for the journal are adhered to including referencing style.

Reviewer #2: This is an important contribution to the epidemiology of cerebral palsy in low- and middle-income countries. I would like to thank the authors for conducting this v important work. Few minor comments for consideration;

Participants and procedure

1. Please add the case definition of CP used in this study/ the reference regarding the definition adopted/used

2. The small sample size is a major limitation in this study, which authors have added in the limitations correctly

3. The authors mentioned that the children were followed-up by phone/home visit. Was this a continuous follow-up? It would be good to have some information on the interval between two follow-ups

4. Conducting the verbal autopsy is the most challenging part in this study. What was the waiting period to conduct the VAs following confirmation of a death of a child included in this study? for both CP and general population?

5. Was type of CP included as a mortality risk factor? Could not find in the results. If not, its worth exploring.

Analysis

6. The total follow-up period was different for children with CP and children in general population. Please add a justification why the follow-up duration was different? What if there were more child at risk of death/died/lost to follow-up between 31 Dec 2017 and 30 Aug 2019 in general population and how was that addressed?

7. It would have been nicer to see the survival probability of undernourished children (e.g. WFA/ HFA/ WFH <-2 to >= -3SD) in addition to the severely undernourished children

8. The authors might consider elaborating the calculation of MR and MRR a bit more for readers from different fields

9. The authors might consider adding the method that was used to include variables in the "single multivariable Cox proportional hazard model" for example, forward selection/ backward elimination/ entry method etc.

Results

10. In the second line of the results section, please rephrase “9 were non-walkers (GMFCS III-V)” to “nine had GMFCS level III-V”

11. I think it would be “MRs for the two cohorts and the MRRs are shown in Table 1” instead of “MRs and MRRs for the two cohorts are shown in Table 1. Please check.

Age

12. Under the paragraph “Age” in results section, it is not clear which population the authors are referring to in the following statement “The effect of age was significant in this population (log-rank, p<0·0001)”. Is it the age group 2-6 years or 10-18 years or both? Please clarify.

Mortality risk factors

13. Paragraph 1: MR was higher in children with severe impairments (GFMCS IV-V; 8718 deaths per 100 000 person years) than in those with mild impairments (GMFCS I-II; 1305 deaths per 100 000 person years). Was it age-sex adjusted analysis? and was other variables e.g. type of CP, type of impairments etc. were checked for association?

14. Paragraph 4: “In a multivariable model including GMFCS subgroup, associated impairments/ seizures, and severe malnutrition, only GMFCS was significantly associated with an increased risk of death, with a HR of 15·4 (95% CI 1·5–156·3; p=0·021) for children with severe impairments, compared with those with mild impairments (Table 3).” � Was there a confounding role? Evidence from other LMICs indicate Children with severe GMFCS are likely to have poor nutritional status as well as higher number of associated impairments.

Immediate cause of death

15. In the last sentence of the paragraph, the authors stated that, the COD could not be determined in 18 children as they moved to unknown locations. Were they lost to follow-up? In that case were they censored at the last follow-up date? in methodology it was mentioned they were considered to have 36 months follow-up. Because the question rises if those children were lost to follow-up, then it is not known if they are alive or died, which will eventually affect the mortality rate in children from general population.

Alternative, might consider excluding them from the analysis?

Discussion

16. In paragraph 2, line 15: Please remove the word ‘only’ before ‘5 times higher….’

Also please check the reference, the authors did not report MRR, they made a comparison with another study in the discussion. Please correct the statement.

Use primary reference for the statement in line 19.

17. Paragraph 3 line 12, previously WFA, HFA and WFH were used to indicate the indicators. Please make it consistent.

18. Paragraph 3 line 19: Might consider rephrasing the statement “relatively simple intervention” or please clarify with examples of “relatively simple intervention”? In my opinion, tube feeding/ gastronomies should not be considered as "relatively simple interventions" in LMIC settings. The authors have explained correctly that those interventions could lead to adverse outcome rather than benefiting the child if not managed properly specially in LMIC settings.

19. Paragraph 3 line 16 and 17: please add references

20. Paragraph 5 line 5: please add reference

21. Paragraph 6 last line: please add reference if available

Table 4

22. Please elaborate the column 2 and 3 headings in Table 4. If understood correctly, the 2nd column indicates the cause of death among children with CP and the 3rd column indicates the cause of death among children from general population? Please clarify

23. Also please add number of deceased in addition to the percentage while presenting cause of death in Table 4.

24. Would recommend adding a footnote clarifying that the COD presented in row 5-9 are only applicable for children in general population and not for children with CP.

Figure 1 and 2

25. These are really good findings and have important implications. Please add p values.

Supplementary Table 1

26. Would recommend adding the “age at baseline assessment” instead of the “age in 2015”

27. The authors have mentioned GMFCS and associated impairments/ seizures ‘NA’ for three children in the table. Does NA means not available/ missing data? In that case please add a footnote to clarify. Similar comment for the variable ‘severe malnutrition’ of the last child in the table and ‘-‘ in the underlying COD variable.

28. The authors have mentioned “Malaria and Anaemia” as underlying and immediate cause of death for six children. It would be good to add some description of the fatal pathway. Is it untreated malaria leading to haemolytic anaemia? In that case was full blood count done for those fatal cases? if not better to put Malaria as immediate cause of death. Also, please add a statement on how malaria was diagnosed?

6. PLOS authors have the option to publish the peer review history of their article (what does this mean?). If published, this will include your full peer review and any attached files.

Reviewer #1: No

Reviewer #2: **Yes: **Gulam Khandaker

---

## [Author Response · Author response to Decision Letter 0]

26 Nov 2020

REBUTTAL LETTER

Dear Dr Barbara Schumann

Thank you for considering our manuscript and inviting us to submit a revised version of the manuscript. 

We would like to acknowledge the high standard of the comments raised by the referee’s, they are obviously experts with great knowledge of this specific topic. We are very thankful and have tried to comply to most of their suggestions and think this has resulted in considerable clarifications and improvements. 

Please, find below the rebuttal letter that responds to each point raised by the academic editor and the reviewer(s). 

We have uploaded the underlying data at the Swedish National Data Service, DOI: 10.5878/xr97-2a37; SND-ID: 2020-178 ; and provided a link: https://snd.gu.se/en/catalogue/study/preview/b58a1ecd-3f49-4ccd-a855-feab627517b8

We have added a Data Sharing Statement at the end of the manuscript providing this information.

We have used the STROBE checklist for reporting of observational study, and uploaded the checklist.

We have followed the Journal Requirements:

i) uploaded this letter as a separate file labelled 'Response to Reviewers'

ii) uploaded a marked-up copy of the manuscript that highlights changes made to the original version ('Revised Manuscript with Track Changes' 

iii) uploaded an unmarked version without tracked changes ('Manuscript').

In the Manuscript version (without tracked changes) we have reformatted the manuscript meeting PLOS ONE's style requirements according to the templates. Note that we have not done this in the Revised Manuscript with Track Changes facilitating observing alterations of the text compared to the first submission. 

We have included captions for S1 Table at the end of the Manuscript. 

 

Point-to-point responses to the reviewers:

Reviewer #1

Case ascertainment is the greatest limitation of the study, simply because the cases are too few, indicating there may have been some others left out for follow-up. Stigma for cerebral palsy (CP) is rampant in these rural settings of subs-Saharan, and for this reason many children with CP will be hidden without disclosing the status to the census staff.

Reply: We agree that there is a multitude of challenges performing a population based study on children with cerebral palsy in rural Uganda due to the stigma, lack of services, lack of knowledge, and lack of registers. This is probably the reason why there were no population based studies on cerebral palsy in low and middle income countries until recently. Now there are two population based cohorts reported; this one and a study using community key informants from Bangladesh. Yet, in spite of these challenges, we were convinced that in order to increase the awareness of children with CP, and to induce behavioural and social change and offer better services, the first step was to explore the prevalence of CP. We addressed the challenges in the following way:

1. We used the infrastructure of the IM-HDSS where the vital data of all children living in the area is monitored annually, i.e., getting the denominator

2. Making a three stage-screening, in which the first step was to visit all households in the area, and to ask about all children registered in the HDSS data base, and any additional.

3. Prior to determining the screening questions for suspect CP, based on the 10Q, we had a number of focal group discussions including caregivers, ordinary community members, community leaders, teachers and health workers. In these groups we explored the knowledge about children with CP, the terminology used, the reason for children getting CP, where caregivers could ask for help and support.

4. The screening questions were thereafter formulated according to local conditions and language in a neutral way not offending the caregiver. All field workers performing the survey were trained how to behave and how to express themselves in order to minimize that parents were “hiding” their children.

5. In addition, after completing the three stage screening, we did a triangulation in which staff acquainted to the communities contacted key informants in all villages, all community health centres and the Paediatric ward at Iganga Hospital.

In spite of taking all these measures, we can of course not be sure that we have identified all children with CP in the district. An interesting observation was that the team working in the communities were often approached by caregivers of children with disabilities from outside the study area, showing that caregivers were willing to brave the stigma when there was a possibility of a professional assessment. Another positive observation was that when we did the four year follow up, we got information about four new cases. However, it turned out that none of them had been living in the area during the first screening. Thus, no indications that we had missed anyone.

This thorough procedure was described in the first report on these cohorts published in Lancet Global Health (LGH) after review of seven reviewers. In that report we also discussed the challenges raised by the reviewer, including the stigma. This publication, and a second report on the characteristics of the group of children with CP, are referenced multiple times in the text for readers who want more details how the sample was collected. We have now also rewritten the Strengths and Limitations and added these limitations and information.

Many children will probably already have died by the 2015, when this study was set up, which can result in few cases available for follow-up in this study (N=97). Reliability of these followed numbers can be done by computing a prevalence and examining if it compares with expected estimates from low income countries (e.g. 4 per 1000 previously reported in Uganda).

Reply: The reviewer is correct that many children with CP, who would have been between 2-17 year when we made the three stage screening in 2015, had died before that screening was performed. Indeed, we noted this when we analysed the screening results. Already in the first LGH article we showed that the number of children in the cohort (and the prevalence) declined with increasing age, and speculated this was due to premature mortality (thus deaths occurring prior to the screening). This was actually one of the incitements to perform this study on the mortality rate. Noteworthy, the prevalence number of 4 per 1000 reported in Uganda which referee #1 is referring to, originates from this cohort and is accurate for children less than 6 years of age; see the LGH article. Due to the high mortality the prevalence declined with increasing age. Hence, we cannot compare with above mentioned expected estimate, since this estimate was derived from the CP cohort and the general cohort studied in this report. 

Failure to identify children with CP increases their inadvertent inclusion into the comparison group selected from the general population, resulting in higher mortality rates in controls, subsequently reducing the comparative or relative mortality ratios.

Reply: We have taken this into consideration and rewritten Strength and Limitations. 

These low numbers due to failure to exhaustively identify all cases in the population would introduce selection bias, making the interpretation of these results difficult.

Reply: We have taken this into consideration and rewritten Strength and Limitations. 

The mortality rates will increase with duration since onset of CP, so early identification of CP would reduce the gross underestimation which is possible with birth cohorts and registers.

Reply: This is an interesting comment, which we cannot address in this study, since we included children first after 2 years of age. Note, that most CP-registers do not include children below 2 years of age. Importantly, early identification of CP is becoming more and more important in order to initiate early intervention, yet, this is still not general practise in most countries. We have emphasized that this study only includes children above two years of age in Strengths and limitations.

In addition to the 97 children identified in 2015, the authors could have considered other ascertain methods including hospital facilities, and rehabilitation services for children who visited these facilities in 2015.

Reply: We considered these circumstances when we did the screening 2015 and added a triangulation including the facilities and services suggested by the reviewer. Thus the sample of 97 children include 11 children identified via the triangulation. Please, see the first reply above and the LGH article for more details. We have now also rewritten the Strengths and Limitations

The period of follow-up is a relatively short and wouldn’t identify many deaths as would a longer follow up, particularly in the general population, which results in higher mortality ratios when compared with CP cases.

Reply: A longer follow-up would correctly result in more deceased children, yet, the statement is not accurate. We used all-cause death per person observation months as our primary outcome for the two groups. That means we first calculated person observation months per group. A longer follow-up period would thus not only result in more deceased children, but also in more person observation months (denominator).

The follow up frequency isn’t very clear in the methods, as to how many follow up phone calls were made over the period for those who were alive on the initial follow up. Multiple sweeps of follow ups would increase the probability of picking up more deaths. This need to be clarified in the methods, otherwise it can complicate the identification of deaths.

Reply: This was a two-time point assessment for the children in the CP group; the first 2015 and the second 2019. When we performed the follow up after 4 years, we first contacted the family via telephone. For families not responding to the call, the “community mobilizer” (who had met the families 2015 and know where they were living) visited the families. By this method we did not miss any child with CP who had been identified 2015.

We have added “All 97 children with CP who were examined 2015 were identified 2019” as the very first sentence in the result section.

We have also added “who had been identified 2015, were contacted by phone or home visit in September 2019” to indicate that there was only one contact.

Considering children in the general population, the information was achieved from the IM-HDSS, which performs annual surveys of the entire population living in the same defined area as the children with CP. Field workers visit every household within the HDSS and monitor vital data for every person living in these households, including people who were deceased since the last census. When a person, including a child, have deceased, specially trained field workers and health professionals perform verbal audits and determine cause of death according to the standard operational procedure of the HDSS. This procedure is described in this reference: (Kananura RM, Leone T, Nareeba T, Kajungu D, Waiswa P, Gjonca A. Under 10 mortality patterns, risk factors, and mechanisms in low resource settings of Eastern Uganda: An analysis of event history demographic and verbal social autopsy data. PLoS One. 2020;15(6):e0234573) and we have now made additional references to this report in the Methods section.

Severe motor impairments are on the spectrum and part of CP and would rather be considered for stratified analysis rather than risk factors analysis. Analysis of malnutrition as a risk factor is justified.

We have changed accordingly all through the manuscript.

How did the individual levels of malnutrition perform in the risk factor analysis i.e. wasting (weight-for-height z scores), weight for age z scores or stunting (height-for-age z scores) compared to the combined category for any of these? These could differentially influence mortality outcomes.

Reply: We chose to combine related measurements into a few variables that we thought could capture a broader indication of nutritional status, and to avoid testing many variables in a relatively small data set. Following the advice, we have now investigated the contribution of weight-for-age (WFA), height-for-age (HFA), and weight-for-height (WFH) in separate models as suggested by the reviewer. We used three Cox models, each having one of the following explanatory variables weight for age z-scores, height-for-age z scores and weight-for-height z scores. We found that only weight-for-age z score was significant (p=0.023). 

This has now been added to the Methods; Analysis and Results; Malnutrition sections. 

It is worth emphasizing the combined malnutrition wasn’t significant at the multivariate but at the univariable level.

Reply: We have further explored the interaction between malnutrition and GMFCS level in the multivariate model in response to Referee #2: comment 14. We have added this information in Results; Malnutrition.

The role of co-existing conditions such as sensori-neural impairments, cognitive impairments and epilepsy are not explicitly reported, yet these can increase premature mortality. If these co-existing conditions are to be examined, they could be separated from risk factors analysis.

Reply: The following associated impairments and conditions were analysed: visual, hearing, intellectual, and behaviour impairments and the presence of seizures (see Methods; Participants and procedures). A brief description is provided in this manuscript referring to more detailed descriptions in one of the previous publications of this CP cohort (Andrews et al 2020). That study showed that most children had several associated impairments. Comparing each impairment individually did not show any significant differences between deceased and surviving groups. Therefore, the children were divided into two groups: one or fewer impairments/seizures and more than one impairments/seizures. The results are presented in Results; Associated impairments: “MR was higher in children with more than one associated impairments/seizures than in those with one or fewer impairments (Table 2). However, survival probability was not significantly related to the number of associated impairments (log-rank, p=0·213; Table 2, Figure 2).”

Causes of death should compared between those for CP and those for the general population.

The older mean age of death for CP compared to controls needed further discussion, as this could suggest continuing risk of death for CP, or that underlying causes of death are different between these groups. Non-infectious causes of death may continue to pause risk in even older ages, while most infections e.g. malaria are in early years.

Reply: We have extended this paragraph and are now discussing the differences in COD between the CP group and the general population, including age pattern differences. See Discussion; Last paragraph.

More limitations need to be noted in the discussion including some outlined above.

Reply: Thanks for your well-grounded reflections and for challenging us. Part of your concerns were reported in previous publications, but we have now extended the limitations and rewritten the Strength and Limitations section.

Discretionary comments

Ensure the formatting instructions for the journal are adhered to including referencing style.

We have tried to adhere to journal instructions.

Reviewer #2: 

This is an important contribution to the epidemiology of cerebral palsy in low- and middle-income countries. I would like to thank the authors for conducting this v important work. Few minor comments for consideration;

Participants and procedure

1. Please add the case definition of CP used in this study/ the reference regarding the definition adopted/used

Reference to definition according to the Surveillance of CP in Europe has been added.

2. The small sample size is a major limitation in this study, which authors have added in the limitations correctly

Agree

3. The authors mentioned that the children were followed-up by phone/home visit. Was this a continuous follow-up? It would be good to have some information on the interval between two follow-ups

Reply: This was a two-point assessment; the first when the children were identified 2015, respectively a second in September 2019. This has now been added in Participants and procedures. Please, see a more extensive response to a comment from Referee #1.

4. Conducting the verbal autopsy is the most challenging part in this study. What was the waiting period to conduct the VAs following confirmation of a death of a child included in this study? for both CP and general population?

Reply: According to the standard operation of the HDSS, monitoring of deaths in the HDSS population is made through both the routine biennial home visits and death notifications by community based scouts. When a death has been confirmed, specially trained field workers with counselling skills are scheduled to do a verbal autopsy after allowing the family at least four to six weeks of mourning. The HDSS field worker administers the WHO verbal autopsy tool to collect information from the caretaker about the deceased and the circumstances around the event (death). The completed tool is then forwarded to the physicians who review and determine the cause of death. The same procedure was followed for children with CP.

This information has now been added to the Methods; Analysis section. We have also included “Although these interviews were made within some months according to the IM-HDSS procedure, there was a risk of information and recall bias”. in Strengths and Limitations.

5. Was type of CP included as a mortality risk factor? Could not find in the results. If not, its worth exploring.

Reply: We choose to use the GMFCS and associated impairments as indicators of the severity level. We thought it would be difficult to use a large number of categorical variables in such a small sample. We have now followed the advice of the referee and used the following CP type categories (according to SCPE): 1 = Unilateral; 2 = Bilateral; 3 = Dyskinetic; 4 = Ataxic; 5 = Hypotonic; 6 = not specified.

Analyses were performed using 1, 2, and 3 separately (log rank test p=0.53; Cox regression p=0.331, and p=0.380 using Unilateral as reference), respectively;

and 1 vs 2+3 (log rank test p=0.27; cox regression p=0.283).

We don’t think this analysis is worthwhile to include. Indeed, our sample was small and it might be possible to identify differences in a bigger sample; mentioned as a Limitation. 

Analysis

6. The total follow-up period was different for children with CP and children in general population. Please add a justification why the follow-up duration was different? What if there were more child at risk of death/died/lost to follow-up between 31 Dec 2017 and 30 Aug 2019 in general population and how was that addressed?

Reply: The sampling procedure for the two cohorts differed. For the CP cohort it was straight forward from the day of examination 2015 to the day of follow-up 2019. This method was not possible for the general population, where we had to set a start date and an end date for which we could extract data from the HDSS register. Due to practical and logistic reasons and time until the HDSS can produce quality assured data, we choose three years from January 2015 to December 2017, which overlapped the three first years of the follow up of the CP cohort. Since we are using all cause death per person month of observation, the different duration of the observation between the two groups will not affect the mortality rate. We have neither any indications that there was any change in mortality in the general population 2018/2019.

7. It would have been nicer to see the survival probability of undernourished children (e.g. WFA/ HFA/ WFH <-2 to >= -3SD) in addition to the severely undernourished children.

Reply: Since 10 of the 13 deceased children with CP had severe malnutrition (<-3SD), we used this as a discriminating threshold. We have now followed the advice of referee #2, and divided the children in three groups; i.e., i) more than -2SD in any of WFA/HFA/WFH; ii) between -2SD – -3SD; and iii) less than -3SD. The log rank test was p=0.076. Cox regression showed no difference between i) vs ii), and between ii) vs iii). An almost significant difference between the most extreme cases i) vs iii) was found (p=0.055). These results are consistent with what we report in the manuscript but do not add much new information. Due to the relatively small number of subjects we opted to use just two categories for the variable characterizing degree of malnutrition.

8. The authors might consider elaborating the calculation of MR and MRR a bit more for readers from different fields

Reply: We have described MR and MRR in more detail in Methods: Analysis section.

9. The authors might consider adding the method that was used to include variables in the "single multivariable Cox proportional hazard model" for example, forward selection/ backward elimination/ entry method etc.

Reply: We did not select variables. We just did one multivariate model using all the risk variables referred, i.e., GMFCS-level, associated impairments/seizures, severe malnutrition.

Results

10. In the second line of the results section, please rephrase “9 were non-walkers (GMFCS III-V)” to “nine had GMFCS level III-V”

Changed accordingly

11. I think it would be “MRs for the two cohorts and the MRRs are shown in Table 1” instead of “MRs and MRRs for the two cohorts are shown in Table 1. Please check.

Changed accordingly.

Age

12. Under the paragraph “Age” in results section, it is not clear which population the authors are referring to in the following statement “The effect of age was significant in this population (log-rank, p<0·0001)”. Is it the age group 2-6 years or 10-18 years or both? Please clarify.

Clarified.

Mortality risk factors

13. Paragraph 1: MR was higher in children with severe impairments (GFMCS IV-V; 8718 deaths per 100 000 person years) than in those with mild impairments (GMFCS I-II; 1305 deaths per 100 000 person years). Was it age-sex adjusted analysis? and was other variables e.g. type of CP, type of impairments etc. were checked for association?

Reply: The results presented were not age or sex adjusted. We decided not to include sex and age in the analysis since there were no significant effects of sex and age on the mortality rate of the CP cohort. We have now repeated the Cox regression including sex and age as variables. The results remain qualitatively the same. The HR was 7.0 (p=0.008) when comparing children with mild and severe impairments. The difference in risk between mild and moderate impairments was not significant (p=0.17). 

The effect of the other variables was studied in the multivariate model. We have changed the text to clarify these issues in Results; Malnutrition.

14. Paragraph 4: “In a multivariable model including GMFCS subgroup, associated impairments/ seizures, and severe malnutrition, only GMFCS was significantly associated with an increased risk of death, with a HR of 15·4 (95% CI 1·5–156·3; p=0·021) for children with severe impairments, compared with those with mild impairments (Table 3).” � Was there a confounding role? Evidence from other LMICs indicate Children with severe GMFCS are likely to have poor nutritional status as well as higher number of associated impairments.

Reply: In Cox model analysis, using GFMCS alone as explanatory variable, we found a significant higher risk of death when comparing children with severe impairment with those with mild impairments. In a model using severe malnutrition as the single explanatory variable we found it also having an effect on mortality. This indicates that GMFCS and malnutrition might be confounded. We have now checked for independence of the three variables (associated impairments, GMFCS and malnutrition) using a Chi-square test. We found that GMFCS and malnutrition are significantly related, but not the other variables. We have now added this in Results; Malnutrition.

Immediate cause of death

15. In the last sentence of the paragraph, the authors stated that, the COD could not be determined in 18 children as they moved to unknown locations. Were they lost to follow-up? In that case were they censored at the last follow-up date? in methodology it was mentioned they were considered to have 36 months follow-up. Because the question rises if those children were lost to follow-up, then it is not known if they are alive or died, which will eventually affect the mortality rate in children from general population.

Alternative, might consider excluding them from the analysis?

Reply: The death of these 18 deceased children in the general population (for whom we could not determine COD) was confirmed (and thus not missing). However, the COD could not be determined because the caretaker, the one who used to take care of the deceased, was not available or had relocated at the time of the interview. The field workers could not administer the VA tool to capture all the required details about this particular death. That means that the death occurred and was confirmed and recorded by the HDSS team, while the COD could not be determined.

We have now clarified this in Results; Immediate cause of death.

Discussion

16. In paragraph 2, line 15: Please remove the word ‘only’ before ‘5 times higher….’

Changed accordingly

Also please check the reference, the authors did not report MRR, they made a comparison with another study in the discussion. Please correct the statement.

Use primary reference for the statement in line 19.

The statement changed accordingly, and primary reference included.

17. Paragraph 3 line 12, previously WFA, HFA and WFH were used to indicate the indicators. Please make it consistent.

Changed accordingly

18. Paragraph 3 line 19: Might consider rephrasing the statement “relatively simple intervention” or please clarify with examples of “relatively simple intervention”? In my opinion, tube feeding/ gastronomies should not be considered as "relatively simple interventions" in LMIC settings. The authors have explained correctly that those interventions could lead to adverse outcome rather than benefiting the child if not managed properly specially in LMIC settings.

“relatively simple” was omitted

19. Paragraph 3 line 16 and 17: please add references

We have added a sentence and referred to the CP cohort from Bangladesh

20. Paragraph 5 line 5: please add reference

Reply: The proposal “These findings suggest that the age pattern for mortality in children with CP differs from that of the general population, and that many died when approaching school age” is derived from the reference (9), just prior to this sentence. We don’t think it needs to be repeated. 

21. Paragraph 6 last line: please add reference if available

We have added three references (two new) 

Table 4

22. Please elaborate the column 2 and 3 headings in Table 4. If understood correctly, the 2nd column indicates the cause of death among children with CP and the 3rd column indicates the cause of death among children from general population? Please clarify.

Clarified.

23. Also please add number of deceased in addition to the percentage while presenting cause of death in Table 4. REMAKE

Remade

24. Would recommend adding a footnote clarifying that the COD presented in row 5-9 are only applicable for children in general population and not for children with CP.

Added in the legend

Figure 1 and 2

25. These are really good findings and have important implications. Please add p values.

p values added to figure legends

Supplementary Table 1

26. Would recommend adding the “age at baseline assessment” instead of the “age in 2015”. REMAKE

Table remade

27. The authors have mentioned GMFCS and associated impairments/ seizures ‘NA’ for three children in the table. Does NA means not available/ missing data? In that case please add a footnote to clarify. Similar comment for the variable ‘severe malnutrition’ of the last child in the table and ‘-‘ in the underlying COD variable.

NA and “-“ have been clarified in the legend.

28. The authors have mentioned “Malaria and Anaemia” as underlying and immediate cause of death for six children. It would be good to add some description of the fatal pathway. Is it untreated malaria leading to haemolytic anaemia? In that case was full blood count done for those fatal cases? if not better to put Malaria as immediate cause of death. Also, please add a statement on how malaria was diagnosed? 

Reply: We are following the validated procedure of the HDSS, and cannot change the immediate COD, nor the underlying COD. The WHO tool used asks for both the immediate and underlying cause of death. The interviews are conducted with the caretaker and the physician who reviews the information determines immediate COD and underlying cause. In these particular cases, the caretaker reported the symptoms and condition of the child that lead the physician to conclude that the COD was anaemia. Further reviews showed that the child also had malaria (either the caretaker had been told at a health facility or by reviewing the symptoms). At health facilities, the main diagnosis techniques for malaria are rapid diagnostic tests and sometimes presumptive diagnosis. However, by the time the caretaker is interviewed, they can only talk about the symptoms and information from health workers but not how the diagnosis was done. There was no full blood count done, and no malaria diagnostics on fatal cases.

We have now rewritten this paragraph of the Discussion to explain in more detail. 

The lack of laboratory tests for anaemia and malaria has been added as a limitation in

---

## [Decision Letter · Decision Letter 1]

1 Dec 2020

Excessive premature mortality among children with cerebral palsy in rural Uganda: a longitudinal, population-based study

PONE-D-20-31617R1

Dear Dr. Forssberg,

We’re pleased to inform you that your manuscript has been judged scientifically suitable for publication and will be formally accepted for publication once it meets all outstanding technical requirements.

Kind regards,

Barbara Schumann, Ph.D.

Academic Editor

PLOS ONE

Additional Editor Comments (optional):

Reviewers' comments:

Reviewer's Responses to Questions

**Comments to the Author**

1. If the authors have adequately addressed your comments raised in a previous round of review and you feel that this manuscript is now acceptable for publication, you may indicate that here to bypass the “Comments to the Author” section, enter your conflict of interest statement in the “Confidential to Editor” section, and submit your "Accept" recommendation.

Reviewer #1: All comments have been addressed

Reviewer #2: All comments have been addressed

2. Is the manuscript technically sound, and do the data support the conclusions?

Reviewer #1: Yes

Reviewer #2: (No Response)

3. Has the statistical analysis been performed appropriately and rigorously? 

Reviewer #1: Yes

Reviewer #2: Yes

4. Have the authors made all data underlying the findings in their manuscript fully available?

Reviewer #1: Yes

Reviewer #2: Yes

5. Is the manuscript presented in an intelligible fashion and written in standard English?

Reviewer #1: Yes

Reviewer #2: Yes

6. Review Comments to the Author

Reviewer #1: The authors have replied to the concerns to the best of their ability; the subject is important and adds to the limited literature in low- and middle-income countries.

Reviewer #2: Thank you for addressing all the comments. This would be an important contribution to the epidemiology of cerebra palsy in low and middle income countries.

7. PLOS authors have the option to publish the peer review history of their article (what does this mean?). If published, this will include your full peer review and any attached files.

Reviewer #1: No

Reviewer #2: **Yes: **Gulam Khandaker

---

## [Editor Report · Acceptance letter]

14 Dec 2020

PONE-D-20-31617R1 

Excessive premature mortality among children with cerebral palsy in rural Uganda: a longitudinal, population-based study 

Dear Dr. Forssberg:

I'm pleased to inform you that your manuscript has been deemed suitable for publication in PLOS ONE. Congratulations! Your manuscript is now with our production department. 

Kind regards, 

on behalf of

Dr. Barbara Schumann 

Academic Editor

PLOS ONE